# Photoinduced modulation of the oxidation state of dibenzothiophene *S*-oxide molecules on an insulating substrate

Mélissa Hankache[1,5], Valentin Magné[2,3,5], Elie Geagea[1], Pablo Simón Marqués [3], Sylvain Clair[1], Luca Giovanelli[1], Christian Loppacher [1], Emmanuel Fodeke[1], Sonia Mallet-Ladeira[4], Eddy Maerten [2], Claire Kammerer [3] ✉, David Madec [2] & Laurent Nony [1] ✉

On-surface chemistry aims to overcome the limitations of conventional in-solution synthesis by taking advantage of the confinement in two dimensions to master highly ordered covalent structures with tailored properties. So far, most of the reported work is conducted on metal substrates and relies on unconventional mechanisms, thereby precluding a direct transposition of well-established organic reactions from solutions to surfaces. In addition, the intrinsic properties and reactivity of metal substrates often limit the activation methods available to trigger on-surface reactions, and photoinduced processes are especially difficult to handle due to quenching of the adsorbed precursor molecules. Herein, the photoinduced deoxygenation of dibenzothiophene *S*-oxide (DBTO) derivatives is transposed from solutions to insulating alkali halide surfaces in ultra-high vacuum. By combining in-solution and on-surface investigations by means of scanning tunneling microscopy, non-contact atomic force microscopy, as well as bias spectroscopy measurements, we provide evidence of the successful on-surface deoxygenation of individual DBTO derivatives under UV irradiation. The photoinduced deoxygenation is conducted at low temperature (<25 K) on a NaCl thin film formed on a Au(111) substrate to yield the reduced dibenzothiophene (DBT) product with excellent chemoselectivity. This work thus opens the way to in-situ photocontrolled charge state manipulation in purely organic compounds.

The synthetic chemists' toolbox includes nowadays a large array of chemical reactions, allowing for virtually any structural or functional modification of organic and metal-organic scaffolds, with high levels of efficiency and selectivity. Highly elaborated molecular structures displaying well-defined properties can thus be prepared from elementary building-blocks by multistep synthesis. However, in-solution chemistry techniques are not suited for the synthesis of poorly soluble π-conjugated extended molecular structures and for the controlled growth of 1D- and 2D-covalent networks, which are key to progress in nanotechnology. In this context, on-surface synthesis has witnessed tremendous development over the last two decades as an alternative bottom-up approach to produce new (metal-)organic nanomaterials and extended molecular structures exhibiting tailored structural, chemical, optical, electronic and magnetic properties[1,2].

Following 2007's Grill, Hecht et al. pioneering work on the formation of covalently bound porphyrinic networks through thermal

[1]Aix Marseille Univ, Université de Toulon, CNRS, IM2NP Marseille, France. [2]LHFA, Université de Toulouse, CNRS, UMR 5069 Toulouse, France. [3]CEMES, Université de Toulouse, CNRS, Toulouse, France. [4]ICT, Université de Toulouse, UAR 2599 Toulouse, France. [5]These authors contributed equally: Mélissa Hankache, Valentin Magné. ✉e-mail: claire.kammerer@cemes.fr; laurent.nony@im2np.fr

activation of halogenated tectons[3], numerous surface-confined reactions have been studied, mostly on well-defined crystalline metallic substrates. To date, knowledge has thus been gathered to master covalent assemblies and intramolecular scaffold modulations leading to well-defined nano-objects of various dimensions and structures[1]. Thermal annealing is typically used to trigger these processes, but electron beam irradiation[4], Scanning Tunneling Microscopy (STM) tip manipulations[4,5], or light irradiation can also be used occasionally[1,6,7]. The choice of the activation method is intimately related to the nature of the underlying surface, which acts as a non-inert support for the 2D-confined reactions, with a major impact on the diffusion, reactivity and optoelectronic properties of the adsorbed species. In particular, in the case of metallic surfaces, the overlap of the precursors' molecular orbitals with the electronic structure of the metal underneath leads to an active role of the substrate, e.g. as a catalyst in thermally induced covalent couplings and as a provider of photoexcited hot electrons in most photoinduced reactions. Consequently, most on-surface reactions have been reported to proceed via unconventional mechanisms, and they are thus not a straightforward transposition of well-known in-solution organic chemistry processes from a 3D- to a 2D-environment.

In this work, in-solution and on-surface investigations were combined to demonstrate that conventional chemical transformations of small molecules can indeed be directly transposed from diluted solutions to surfaces through a careful combination of precursor design, activation method and, most importantly, substrate nature. Insulating substrates were selected here as supports for the 2D-confined reaction, since the electronic decoupling of overlying adsorbates preserves their intrinsic electronic and optical molecular characteristics[8,9]. It was thus hypothesized that, on insulators, the chemical reactivity of molecular precursors could be anticipated, based on their behavior in diluted solutions. Moreover, it also becomes possible to probe in-situ the native properties of the newly formed products and to stabilize charged states on such surfaces, in contrast with metallic ones.

In spite of these major assets, synthesis on insulating substrates is still in its infancy[10], with only scarce reports to date of successful reactions on bulk insulators such as calcite or alkali halides[11–13]. The key challenges on such substrates lie in the absence of inherent catalytic activity, while possibilities for thermal activation are limited by weak molecule-substrate interactions, resulting in enhanced dewetting and desorption of molecular species. These issues may be overcome by combining a strategic molecular design[14], in particular with polar functional groups, and light as an energy source for chemical reactions[6]. In this regard, biarylsulfoxides appear particularly promising, owing to the significant polarity of their S = O sulfinyl group coupled with their well-established photoreactivity in solution. For instance, dibenzothiophene S-oxides (DBTO) are known to undergo a cleavage of the S = O bond under UV-irradiation, thereby releasing atomic triplet oxygen O($^3$P) and dibenzothiophene (DBT)[15–17]. During this reaction, the sulfur atom oxidation state varies from (0) to (-II) and the photodeoxygenation of sulfoxides thus represents a powerful and reagent-free synthetic tool formally allowing to perform the reduction of sulfur.

Herein, we report the direct transposition of an organic reaction, namely the photodeoxygenation, from solutions to insulating alkali halide surfaces under Ultra-High Vacuum (UHV) conditions, to allow for the photocontrolled modulation of the oxidation state of sulfur atoms in-situ. A sulfoxide precursor displaying finely tuned molecular properties was thus designed, synthesized and its photoreactivity in solution was assessed in terms of efficiency and selectivity. In parallel, its adsorption configuration on an alkali halide surface was unveiled by means of Scanning Probe Microscopy (SPM) techniques, as previous investigations of sulfoxides on surfaces were restricted to metallic ones. Finally, deoxygenation of this model compound was successfully achieved on surface upon irradiation, giving rise to the corresponding sulfide product, and the formal in-situ reduction of sulfur was evidenced by bias spectroscopy measurements.

## Results and discussion

### Design, synthesis and characterization of the sulfoxide precursor

The photodeoxygenation of dibenzothiophene S-oxides, leading to the formal reduction of sulfur(0) to sulfur(-II), has been reported in solution on diversely substituted DBTO derivatives[15–17]. In these systems, the dibenzothiophene core exhibits a fully planar geometry, but the pyramidal character of the sulfoxide moiety induces an out-of-plane location of the oxygen atom, thus leading to an overall bent geometry of the molecule. In anticipation of in-depth structural characterization and photoreactions on surface, the design of the DBTO precursor was adjusted to favor its face-on adsorption on weakly interacting insulating substrates.

Compared to metal surfaces on which the adsorption energy of planarly adsorbed aromatic molecules is relatively high due to the interaction between the molecule's π-electrons and the electrons of the metal substrate, on insulating substrates, the fabrication of molecular layers in a planar adsorption geometry is rather challenging. Prior research on insulating surfaces, both theoretical and experimental, has emphasized the role of polar anchoring groups in the development of such highly ordered self-assemblies[18–21]. Although DBTO molecules natively display an S = O sulfinyl group with significant polarity, it was envisioned to insert two additional electron-withdrawing bromine atoms on symmetrical *ortho* positions (with respect to the bisarylsulfoxide moiety), with the aim to optimize the adsorption energy on alkali halide substrates. 4,6-Dibromodibenzo[*b,d*]thiophene 5-oxide ((*o*-Br)$_2$-DBTO, Fig. 1 left) was thus synthesized and fully characterized (see Supplementary Information file). Its structure was unambiguously confirmed by single-crystal X-ray diffraction, revealing an angle of 117° between the plane of the polycyclic scaffold and the S = O bond (Fig. 1 and Fig. S22). In addition, thermogravimetric analysis (TGA) was carried out in anticipation of on-surface physical vapor deposition of this model compound, which highlighted its thermal stability up to ca. 200 °C (Fig. S11).

### In-solution photoreactivity of (*o*-Br)$_2$-DBTO

The absorption spectrum of (*o*-Br)$_2$-DBTO in dichloromethane (DCM) displays two wide absorption bands peaking at 332 and 278 nm, and a main feature in the 260–230 nm range (Fig. S12a). Photoreactivity studies were thus performed under UVA-irradiation (280–365 nm). Noteworthy, wavelengths <280 nm were excluded as this leads to the cleavage of the bromine anchoring groups as a side reaction. On the first hand, the photoreactivity of (*o*-Br)$_2$-DBTO was investigated in a dichloromethane solution at room temperature (RT). After 48 h of UVA-irradiation, the expected deoxygenated and fully planar compound (*o*-Br)$_2$-DBT was obtained as the major product in 47% isolated yield (Fig. 1). Besides, a byproduct incorporating one additional oxygen atom was isolated in 16% yield. Its structure was unambiguously assigned as the sultine Br$_2$-OTO thanks to an X-ray crystal structure, which revealed the presence of the oxygen atom embedded in the central six-membered ring (Fig. S23). Analogous sultine byproducts had already been evidenced upon in-solution photodeoxygenation of DBTO derivatives[16]. It is important to note that the photoreaction of (*o*-Br)$_2$-DBTO was run on a thoroughly deoxygenated solution, with the dichloromethane solvent degassed by several freeze-pump-thaw cycles. With less rigorous conditions, e.g. by simple argon sparging, a small amount of the sulfone (*o*-Br)$_2$-DBTO$_2$ as additional oxidation byproduct was observed (see Supplementary Information file), presumably due to side reactions involving adventitious molecular oxygen.

From a mechanistic point of view, upon UVA-irradiation, the sulfur(0) precursor (*o*-Br)$_2$-DBTO undergoes photodeoxygenation to give rise to the reduced sulfur(-II) product (*o*-Br)$_2$-DBT and releases atomic triplet oxygen O($^3$P) in the reaction medium[15]. The latter may then be trapped by remaining (*o*-Br)$_2$-DBTO molecules, leading to the

**Fig. 1 | In-solution photoreactivity of (o-Br)₂-DBTO sulfoxide.** Upon UVA (280 – 365 nm) irradiation of the **(o-Br)₂-DBTO** precursor in DCM for 48 h at RT, the reduced sulfur(-II) derivative **(o-Br)₂-DBT** was obtained as major product in 47% isolated yield, along with oxidized sulfur(+II) compound **Br₂-OTO** (16%). The formal oxidation state of the sulfur atom in each compound is given above the reaction scheme. ORTEP side views of the molecular structure of **(o-Br)₂-DBTO** and **(o-Br)₂-DBT**[50] are given below the corresponding chemical structure (thermal ellipsoids drawn at 50% probability; the C, H, O, S, and Br atoms are represented in gray, white, red, yellow, and orange, respectively).

formation of the oxidized sulfur(+II) byproduct **Br₂-OTO**, presumably upon C-S α-cleavage[22]. Consequently, in solution, the efficiency of the photodeoxygenation reaction is hampered by intermolecular oxygen transfers, which inherently limit the selectivity of this process and the control of the resulting sulfur oxidation state.

Once the model compound designed, synthesized and its in-solution photoreactivity assessed, the next step was the investigation of **(o-Br)₂-DBTO** sulfoxides on surface. Surprisingly, SPM studies involving sulfinyl derivatives on surface are scarce and restricted to metallic substrates, mainly involving dimethylsulfoxide (DMSO) and tetramethylenesulfoxide[23,24]. A recent investigation also examined the impact of the molecular configurations of sulfonyl chloride compounds on desulfonylation reactions occurring on Au(111) and Ag(111) surfaces[25].

### NaCl(100) thin films / Au(111) substrate

Beyond the molecule's structure and the presence of polar anchoring groups, the adsorption of precursors on alkali halide substrates is critically influenced by the crystal's ionic structure[20,21,26–28]. Indeed, matching the molecular network, i.e. the position of its electronegative anchoring groups, with the cationic lattice is crucial to control the self-assembly on ionic substrates. Therefore, the alkali halide surface NaCl(100) was chosen, since it features a cation distance (nearest neighbor: $d_{Na-Na} = 0.4$ nm) matching quite well the Br−O distance ($d_{Br-O} = 0.36$ nm) and, to a less extent, the Br−Br distance ($d_{Br-Br} = 0.66$ nm) of the functional groups in the **(o-Br)₂-DBTO** molecular tecton. Satisfyingly, upon deposition on bulk NaCl(100) at room temperature, **(o-Br)₂-DBTO** molecules adsorbed in a planar geometry and formed stable monolayers (ML), with each polar functional group potentially located on a cationic site (Fig. 2d).

In this work, the main challenge lies in establishing the proof of a photocontrolled modulation of the oxidation state of sulfur-based molecules by high-resolution SPM, using functionalized CO-tips to image the planarly adsorbed molecules[29]. Since our aim is to characterize both structural (oxygen loss) and electronic modifications (change in sulfur oxidation state) upon irradiation, ultrathin insulating layers were deposited on a metal substrate to allow us to conduct STM measurements in combination with non-contact-Atomic Force Microscopy (nc-AFM) measurements (hereafter referred to as dynamic STM, see "Methods"). Indeed, whereas STM cannot be performed on bulk insulating crystals, electron tunneling is transparent through ultrathin layers of wide-bandgap insulators (one to three monolayers), such as oxides or alkali halides, deposited on metal surfaces[30,31]. Additionally, it

has been demonstrated that in the case of ultrathin NaCl films, one double layer already exhibits the distinctive electronic structure of its bulk form[32,33], meaning that the NaCl layers on metals are thin enough to allow electrons to tunnel while preserving the optical properties of the adsorbed molecules.

NaCl islands have been successfully grown on a wide range of crystalline metal surfaces, including Ge(100)[30], Al(111)[31,34], Al(100)[31,34], Cu(111)[33], Au(111)[35,36], Ag(100)[37], Ag(001)[38] and Ag(110)[39]. According to these experimental investigations and numerous theoretical studies, NaCl forms (100) oriented atomic layers on these substrates. In the frame of this study, NaCl thin layers were deposited on gold surface Au(111) (see "Methods" and Supplementary Information file). As expected, STM images indicate that NaCl grows as (100) layers on the reconstructed (111) gold terraces. A high-resolution STM image of the NaCl unit cell (Fig. S2c) shows that its lattice constant of 5.65 Å is consistent with that of the NaCl bulk crystal. Larger-scale STM images (Fig. S2a) reveal a carpetlike growth of the initial NaCl(100) layer, with the formation of the second and third NaCl layers on the first one before the latter entirely covers the Au surface.

The NaCl layers on Au(111) form an ideal substrate for the present study, since they display an atomically clean surface on which (i) the adsorption of the **(o-Br)₂-DBTO** precursor molecules is expected to be in a planar geometry, (ii) such molecules will be electronically decoupled from the metal substrate, thus preserving their intrinsic photophysical properties and allowing UV-induced deoxygenation, and (iii) high-resolution SPM imaging with functionalized CO-tips is possible to observe the expected chemical modification of the molecules and characterize the change in the sulfur atom oxidation state.

### Description of the supramolecular phase

A typical STM image following the deposition of 0.1 ML of **(o-Br)₂-DBTO** on two monolayers (2 ML) of NaCl(100) is displayed in Fig. 2a. It appears that the molecular islands, denoted by the white arrows, are substantially longer than they are wide, suggesting that in-line interactions are stronger than lateral ones. Additionally, the image shows that the molecules preferentially nucleate in the NaCl's polar directions [011] and [01$\bar{1}$]. This is more visible in Fig.2b where the molecular rows of two 90°-oriented domains are resolved along with the atomic-scale contrast of a neighboring third layer of NaCl. A close inspection on a zoomed STM image (Fig. 2c) revealed that the molecules self-assemble into parallel stripes, consisting of oppositely oriented molecular rows. The inter-rows distance is ~12 Å, whereas the distance between two consecutive molecules within the same row is ~8 Å. These

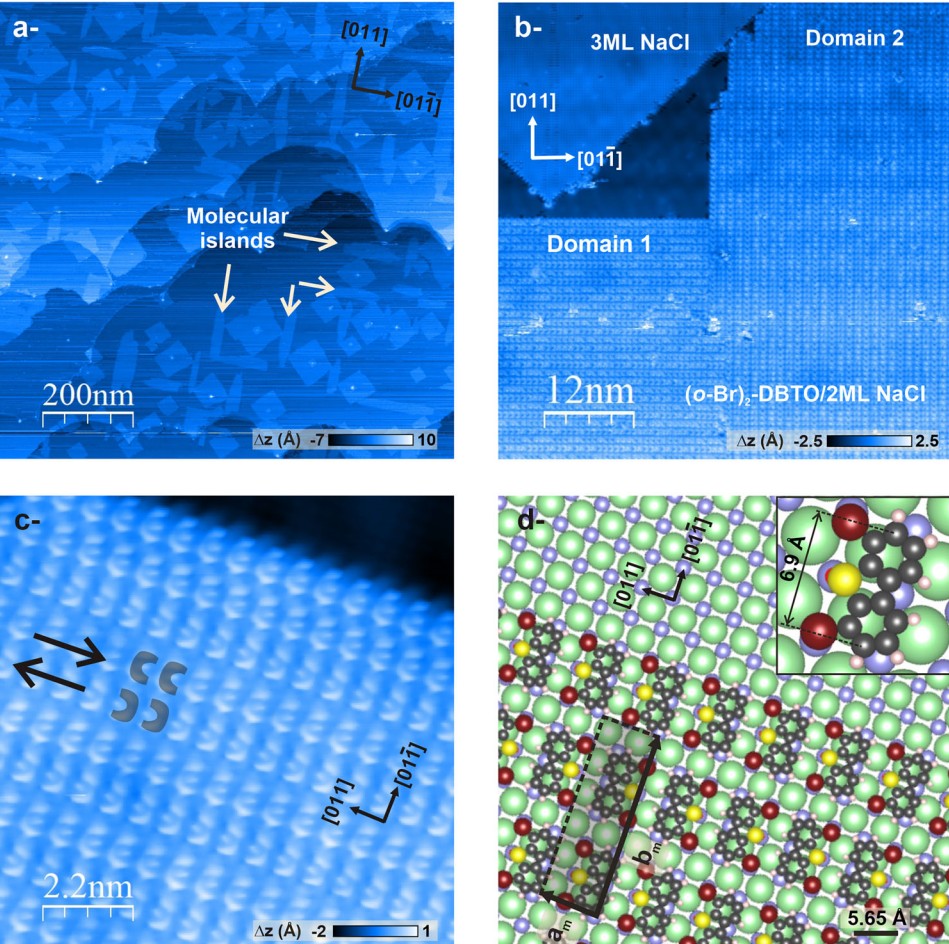

**Fig. 2 | Evaluation of the self-assembly structure of (o-Br)₂-DBTO molecules on 2 ML NaCl(100)/Au(111).** Large-scale (**a**, **b**) and zoomed (**c**) STM images were taken after the deposition of ~ 0.1 ML of (**o-Br**)₂-**DBTO** onto a thin layer of NaCl(100)/Au(111) kept at RT. **a** The molecular islands, depicted with white arrows, preferentially nucleate in the NaCl's polar directions [011] and [01$\bar{1}$]. **b** The growth direction of the molecular islands is better evidenced with these two 90°-oriented equivalent domains near a third layer of NaCl imaged with atomic resolution. **c** Close-up showing the two oppositely oriented rows **of** (**o-Br**)₂-**DBTO** molecules. Four molecules in two opposite orientations have been overlaid with a gray croissant shape. STM parameters are (**a**) $I_t$ = 12 pA; $V_b$ = −1.9 V, (**b**) $I_t$ = 8.5 pA;

$V_b$ = 1.1 V and (**c**) $I_t$ = 20 pA; $V_b$ = 1.2 V. **d** Structural model for the molecular unit cell. The C, H, O, S, and Br atoms are represented by gray, white, red, yellow, and burgundy spheres, respectively. The NaCl structure is presented by purple (Na⁺ cations) and green (Cl⁻ anions) spheres. The gray rectangular overlayer depicts the molecular unit cell. The latter is represented by ($a_m$ x $b_m$). Inset: single (**o-Br**)₂-**DBTO** molecule placed on NaCl with an adsorption arrangement illustrating the good fit of the molecular structure with the substrate, with a point-on-point arrangement of the electronegative Br and O anchoring atoms with respect to sodium cations.

values are coherent with a ($a_m \times b_m$ = 6$a$ × 2$a$) rectangular molecular unit cell consisting of 2 molecules, where parameter $a$ = 5.65/√2 ~ 4 Å represents the distance between the nearest cations along the [011] directions in a NaCl crystal (polar directions). Figure 2d presents a structural model for this self-assembly, which illustrates a point-on-point adsorption, highlighting the match between $d_{Na-Na}$ and $d_{Br-Br}$, as anticipated. Within a molecular row, (**o-Br**)₂-**DBTO** molecules adsorb every second line of cations, while the electronegative atoms (both bromine atoms and the oxygen of the sulfinyl group, see below) match the cationic sites. We may therefore conclude that the growth of these lines is driven by the adsorption of the anchoring groups matching the periodicity of the substrate and stabilized by van der Waals (vdW) interactions. Residual electrostatic interactions are optimized by alternating the molecular orientation in neighboring lines while maintaining a rectangular unit cell. As evidenced by the work of Neff et al. for one dimensional lines of vertically adsorbed molecules[40], the parallel aligned dipole moments are sufficiently large to produce a remarkable long-range repulsion between molecular lines on an insulating substrate. In the present case, the molecules are planarly adsorbed with their dipole moment essentially aligned parallel to the

surface (Fig. S8b). We assume that the oppositely oriented lines of molecules are a result of maximized (attractive) electrostatic interactions between the intrinsic dipoles of the molecules.

Finally, the adsorption configuration of each single molecule within the supramolecular assembly was investigated, as the pyramidal shape of the sulfinyl group (Fig. 1, bottom left) leads to two scenarios for (**o-Br**)₂-**DBTO** on the surface. Indeed, the molecule may adsorb with its polycyclic scaffold parallel to the surface and the out-of-plane oxygen atom pointing upwards, or towards the surface. To unambiguously determine the configuration of (**o-Br**)₂-**DBTO** on NaCl thin layers, the Probe Particle Model (PPM), a numerical AFM model that simulates high-resolution non-contact AFM images with CO-functionalized tips[41,42], was used. Simple probe particle approaches combined with experimental measurements including dibenzothiophene have already been shown to be effective in providing rapid structural assignment[43].

High-resolution nc-AFM images of (**o-Br**)₂-**DBTO** issued from experimental observations with a CO-functionalized tip (Fig. S3b) were compared with those obtained from PPM simulations (Fig. S1b,c). The calculated images show that an oxygen-up configuration (Fig. S1c)

would result in a very distinct contrast compared to a molecule with its oxygen facing the surface (Fig. S1b). Based on experimental images (Fig. S3b), which display a weak contrast in the central part of the molecules, it was thus inferred that (**o-Br**)$_2$**-DBTO** is adsorbed with the oxygen pointing towards the substrate. This can occur through the favorable interaction between the oxygen and the cations of the NaCl thin layer (Fig. 2d)[11]. As a consequence, the sulfur atoms of (**o-Br**)$_2$**-DBTO** molecules are spot on the bright circle-shaped protrusions visible for each molecule (Fig. 2c). Importantly, in anticipation of photoreactivity studies, PPM-simulated images show very limited differences in the imaging contrast between a (**o-Br**)$_2$**-DBTO** molecule with the oxygen pointing down (Fig. S1b) and a deoxygenated one, i.e. (**o-Br**)$_2$**-DBT** (Fig. S1a). It is also worth noting that in these simulations, the carbon scaffold of all molecules was kept parallel to the substrate surface. In experimental conditions, this is expected for the planar $C_{2v}$ symmetric (**o-Br**)$_2$**-DBT** molecule, whereas the adsorption of the bent (**o-Br**)$_2$**-DBTO** sulfoxide derivative with the oxygen facing the surface is likely to be slightly tilted with respect to the surface. This induces a modification of the molecular contrast, allowing a clear distinction between the pristine (**o-Br**)$_2$**-DBTO** and the deoxygenated (**o-Br**)$_2$**-DBT** molecules (see below).

## On-surface photoinduced deoxygenation

Once the self-assembly of (**o-Br**)$_2$**-DBTO** molecules thoroughly characterized, their on-surface photoreactivity was explored by both, STM imaging and Local Contact Potential Difference (LCPD) measurements, as deduced from bias spectroscopy (see below). The surface was thus exposed to a 280 nm wavelength irradiation, in line with in-solution photochemistry experiments. This was performed by using a light emitting diode (LED) and irradiating the sample inside the SPM head (see "Methods").

Following illumination, STM imaging in gentle conditions indicate that the molecules now exhibit two distinct contrasts, which is indicative of on-surface deoxygenation. Figure 3a displays a molecular island after 270 min of light exposure. The sulfur atoms of (**o-Br**)$_2$**-DBTO** molecules are still identifiable as bright circle-shaped protrusions, however now, the molecules exhibit a croissant-like shape and a butterfly-like shape (see corresponding gray forms in Fig. 3a). Importantly, it is noticed that the illumination and the subsequent structural modification of some molecules does not change the unit cell of the self-assembly (Fig. 3b). To assign a particular contrast to the pristine

sulfoxide, or to the deoxygenated compound, we performed high-resolution dynamic STM imaging of the molecular assembly using a CO-functionalized tip (Fig. S3a). In that figure, the deoxygenated (**o-Br**)$_2$**-DBT** molecules are identified with white frames. The associated frequency shift map (Fig. S3b) shows only a minor change between (**o-Br**)$_2$**-DBTO** and (**o-Br**)$_2$**-DBT** molecules, as already expected from the series of PPM simulations (Fig. S1a,b). A slight modification of the contrast is noticeable: the tip-molecule interaction is no longer as repulsive above the sulfur atom. As explained earlier, this is rather an effect of a modified adsorption geometry of the now planar, deoxygenated (**o-Br**)$_2$**-DBT** molecule compared to its tilted precursor, than a direct visualization of the missing oxygen atom. Therefore, these observations allow us to unambiguously assign in Fig. 3a the croissant-like shaped molecules to the pristine (**o-Br**)$_2$**-DBTO** sulfoxides and the butterfly-like shaped ones to the (**o-Br**)$_2$**-DBT** deoxygenated sulfoxides. Importantly, the **Br$_2$-OTO** sultine byproduct evidenced upon in-solution photoreactions has never been observed on surface after illumination.

We then carried out a statistical study of the deoxygenation yield (deoxygenated molecules / intact molecules) of molecules on the surface as a function of sample illumination time. After each illumination phase, several images of around 450 molecules each were acquired from different areas of the sample, and the deoxygenation yield was estimated for each image by two separate methods, both of which gave the same yield estimate (Fig. S5 and S6). The deoxygenation yield increases rapidly with illumination time (0–30% in 50 min), then reaches saturation at 40%, after >6 h of illumination (Fig. S4). The underlying cause of this saturation remains to be investigated. The influence of the pristine/deoxygenated state of molecules close to a given molecule on its probability of deoxygenation has also been investigated (Fig. S7). Statistical analysis tends to show that the oxidation state of molecules close to a given molecule (all deoxygenated, none deoxygenated, or a mixture of both) has little influence on its ability to undergo photodeoxygenation. Along this line, it was neither observed that the Au(111) reconstruction, visible underneath the 2 ML NaCl with our tunneling conditions, had an influence on the location of the deoxygenated molecules. The deoxygenation process therefore seems to occur essentially randomly on the surface, whatever the adsorption spot and the neighboring of the molecules.

To complete this structural characterization and analyze the charge state of sulfur atoms before and after deoxygenation, two types

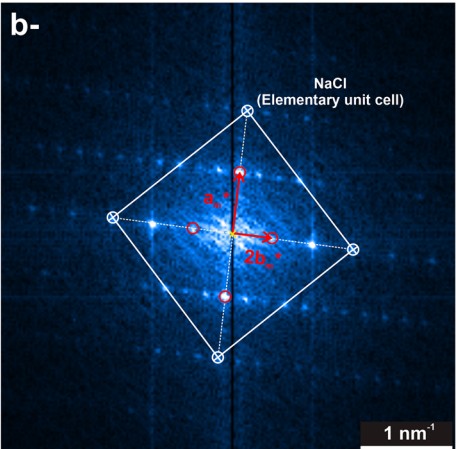

**Fig. 3 | Influence of UV-irradiation on the self-assembly of molecules. a** STM image ($I_t$ = 6.3 pA; $V_b$ = 1.1 V) of the molecular self-assembly after 270 min exposure to the 280 nm LED. The sample was held at low temperature (25 K) during illumination. Gray overlays were used to indicate the two different structures of molecules, namely butterfly-like for deoxygenated (**o-Br**)$_2$**-DBT** molecules and croissant-like for pristine (**o-Br**)$_2$**-DBTO** molecules. **b** Corresponding fast Fourier transform

(FFT). The NaCl unit cell is visible and so are the reciprocal molecular unit cell vectors (m subscripts). Note that due to the image contrast, the FFT is more sensitive to the $2\mathbf{b_m}^*$ vector rather than to the $\mathbf{b_m}^*$ one. Nevertheless, the ($a_m \times b_m = 6a \times 2a$) epitaxial relationship between the molecular unit cell and that of the NaCl can unambiguously be established.

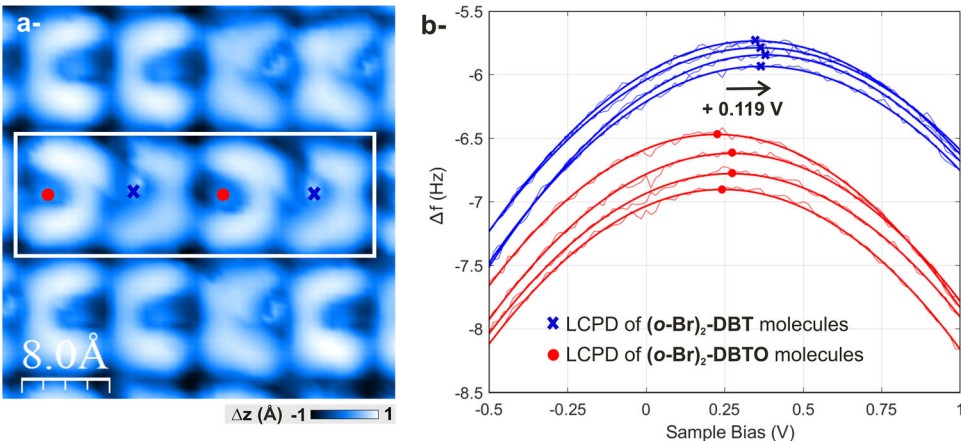

**Fig. 4 | Bias spectroscopy measurements. a** Dynamic STM image ($< I_t > = 2.4$ pA; $V_b = 0.85$ V and $A = 100$ pm) of a ($3 \times 4$) grid of molecules after irradiation with UV light (280 nm). **b** Bias spectroscopy curves $\Delta f(V)$ performed on top of the sulfur atom of each molecule in the white inset shown in image (**a**). For each curve, a parabolic fit and corresponding peak ($V^*$, $\Delta f^*$) are shown. The latter are indicated by the blue x and red • symbols, for deoxygenated (**o-Br**)$_2$-**DBT** and pristine (**o-Br**)$_2$-**DBTO** molecules, respectively.

of bias spectroscopy experiments were performed. On the one hand, individual molecules were selected among a statistical set of pristine/deoxygenated species. A series of $\Delta f$ vs. bias voltage curves ($\Delta f(V)$, so-called bias spectroscopy curves) were performed on top of the sulfur atom of each of these molecules. Bias spectroscopy curves are known to exhibit a typical parabolic shape, whose maximum ($V^*$, $\Delta f^*$) is a measurement of the Local Contact Potential Difference (LCPD), a quantity that can be connected to the local work function difference between the tip and the surface. When the bias voltage matches $V^*$, the local electrostatic forces occurring between the tip and the surface are canceled out. Any charge state manipulation of an atom on the surface (tip manipulation, bias pulses, light…) induces a change of its LCPD, which can therefore be tracked by bias spectroscopy. This was demonstrated in 2009 by L. Gross et al.[44] using a qPlus operated at low temperature, which is like our experimental conditions. They have shown that a negatively charged single Au atom adsorbed on an ultrathin NaCl film on Cu(111) featured a larger LCPD than its neutral counterpart. On the other hand, bias spectroscopy curves were recorded on preset grids, and the $V^*$ was then mapped in two dimensions above the corresponding area. Both point- and grid-spectroscopy experiments yield the same results. However, to keep this text concise, we only report here the results obtained with point-spectroscopy. The grid spectroscopy results are reported in Fig. S10.

Before drawing any conclusions from our bias spectroscopy study, the charge distributions of the pristine and deoxygenated molecules must be compared. To this end, DFT calculations with Natural Bond Orbital (NBO) analysis were undertaken (see "Methods" and Supplementary Information file), showing that the sulfur atom in the deoxygenated (**o-Br**)$_2$-**DBT** molecule displays a less positive partial charge than its pristine counterpart (Fig. S8). Therefore, as stated by Gross et al., we anticipate that, upon further measurement, the experimental spectra corresponding to the deoxygenated molecules will shift to more positive values when compared to those of the pristine (**o-Br**)$_2$-**DBTO** molecules.

Figure 4a shows one of the molecular matrices on which the bias spectroscopy was performed. Three horizontal rows, each containing four molecules, are visible. Two bias spectroscopy curves were conducted on each molecule, with a total of eight measurements per line. The spectroscopy curves corresponding to the four molecules in the white frame are presented in Fig. 4b (red • and blue x correspond to the croissant-like pristine (**o-Br**)$_2$-**DBTO** and butterfly-like deoxygenated (**o-Br**)$_2$-**DBT** molecules, respectively). Experimentally, the tip was placed above a sulfur atom and a $\Delta f(V)$ curve was measured for each

molecule. STM images were taken before and after each series of four curves to confirm that the molecules did not change position. Next, a parabolic fit was performed on the gathered data. In light of this, a global examination reveals that the bias spectra of the deoxygenated (**o-Br**)$_2$-**DBT** molecules (blue x) are consistently shifted to the right with respect to those of the pristine (**o-Br**)$_2$-**DBTO** molecules (red •). The average LCPD shift of S with respect to S=O is found to be (0.119 ± 0.045) V (see Fig. S9 for detailed explanations and a full bias spectroscopy study of 24 molecules). Bias spectroscopy experiments then confirm that the charge state of the sulfur atom is directly impacted by its chemical environment, and more particularly by the presence of the electronegative oxygen atom, with the sulfur displaying a less positive charge state in the deoxygenated (**o-Br**)$_2$-**DBT** molecules. Most importantly, owing to the decoupling character of the NaCl thin layer, such difference in the charge state of sulfur atoms in parent dibenzothiophene derivatives such as (**o-Br**)$_2$-**DBT** and (**o-Br**)$_2$-**DBTO** can be directly probed on surface, which opens the way to in-situ charge state manipulation in purely organic compounds. In the present case, the photoinduced modulation of the charge state of a sulfur atom was successfully achieved on surface.

Finally, it is important to note that the photoreactivity of (**o-Br**)$_2$-**DBTO** sulfoxide on surface under Ultra-High Vacuum features excellent chemoselectivity, since the sultine byproduct **Br$_2$-OTO**, resulting from intermolecular oxygen transfer, was never observed. This is a striking difference from the in-solution reaction, which encourages the transposition of further photochemical reactions from solutions to surfaces.

We have presented a comprehensive on-surface study of a specially designed dibenzothiophene $S$-oxide molecule adsorbed on an ionic substrate, including the characterization of this sulfoxide's adsorption configuration and photoreactivity on such surfaces. Polar anchoring groups were strategically added to the molecular scaffold to match up to the substrate's cationic lattice, resulting in planar adsorption and the formation of an ordered molecular self-assembly arrangement with oppositely oriented rows. Since the adsorbed molecules interact very weakly with both, the underlying ionic substrate and the neighboring species, the (**o-Br**)$_2$-**DBTO** model compound appeared as the ideal candidate to directly transpose the photoinduced deoxygenation process from solutions to surfaces. Using UV-light with similar energy, both in-solution and on-surface reactions induced a modulation of the oxidation state of the sulfur atom in (**o-Br**)$_2$-**DBTO**, although with different levels of efficiency and chemoselectivity. Indeed, the on-surface reaction resulted in a

successful photoinduced deoxygenation of **(o-Br)₂-DBTO** on crystalline NaCl thin films on Au(111) at low temperature (<25 K), yielding selectively the reduced **(o-Br)₂-DBT** sulfide. The first indication of the deoxygenated molecules was a slight, noticeable structural alteration seen in the topographical imaging. Local charges were then measured above the sulfur atom to further prove the deoxygenation, and these measurements depicted a shift towards a less positive charge state, in agreement with theoretical calculations.

Our work thus shows that by properly choosing both, the substrate and the precursor molecules, conventional photochemical reactions can directly be transposed from solutions to surfaces. In addition, the successful modulation of sulfur oxidation state upon on-surface photodeoxygenation opens the way to in-situ photocontrolled charge state manipulation in purely organic compounds.

## Methods

### General methods for the synthesis and characterization of molecules

4,6-Dibromodibenzo[*b,d*]thiophene was purchased from BLD-Pharm and used directly without further purification. Technical grade solvents and reagents were purchased from commercial suppliers and used without further purification. The concentration of the *m*-chloroperbenzoic acid (*m*CPBA) batch indicated by the supplier was ≤77%. Dichloromethane (DCM) used in photoreactivity studies was dried by passing over two columns of activated alumina, kept over activated 4 Å molecular sieves, and degassed by thorough argon sparging or by freeze-pump-thaw cycles. All water was deionized before use. 'Room temperature' (RT) can vary between 18 °C and 25 °C. Irradiation of **(o-Br)₂-DBTO** solution was carried using two 15 W UVA fluorescent tubes from Vilber-Lourmat (see Fig. S13). Analytical Thin Layer Chromatography (TLC) was performed on Merck aluminium-backed silica gel 60 F254 plates. Developed TLC plates were visualized by ultraviolet (UV) irradiation (254 nm). Column chromatography was carried out using Merck silica gel 60 Å, 220 - 440 mesh. Fourier Transform Infrared Spectroscopy (FTIR) was carried out using a ThermoNicolet 6700 with an Attenuated Total Reflection (ATR) attachment and peaks are reported in terms of frequency of absorption (cm⁻¹). High Resolution Mass Spectrometry (HRMS) spectra were acquired using a GCT Premier CAB109 TOF mass spectrometer equipped with DCI-CH₄ ionization. HRMS data were quoted to four decimal places (0.1 mDa). All NMR spectra were recorded on either a Bruker AV 300 or Bruker AV 500 and are internally referenced to residual solvent signals (CDCl₃ is referenced at δ 7.26 and 77.16 for ¹H and ¹³C NMR, respectively). All NMR chemical shifts (δ) were reported in parts per million (ppm) and coupling constants (*J*) are given in Hertz (Hz). The ¹H NMR spectra are reported as follows: δ (multiplicity, coupling constant *J*, number of protons). UV-vis absorbance was measured using a Varian Cary 5000.

### Synthesis of 4,6-dibromodibenzo[*b,d*]thiophene 5-oxide ((o-Br)₂-DBTO)

A solution of 4,6-dibromodibenzo[*b,d*]thiophene (1.03 g, 3.0 mmol, 1 equiv.) in dry DCM (20 mL) in a round bottom flask was cooled to -30 °C. BF₃·OEt₂ (3.0 mL, 24 mmol, 8 equiv.) was added dropwise and stirred for 10 min. A suspension of *m*CPBA (741 mg, 3.3 mmol, 1.1 equiv.) in DCM (5 mL) was then added dropwise. The cool bath was removed and the slurry reaction medium was stirred at room temperature for 24 h. Aqueous NaOH (1 M, 50 mL) and DCM (25 mL) were added, after vigorous stirring both phases were separated and the aqueous fraction was extracted once with DCM (25 mL). Combined organic phases were dried using Na₂SO₄, filtered (rinsing the solids with hot DCM thrice) and the volatiles were removed under vacuum. The crude reaction mixture was then dissolved in a minimum amount of boiling dichloromethane, colorless crystals (420 mg) were obtained upon cooling and standing at room temperature overnight. A second crop of crystals (210 mg) could be obtained from further

recrystallization of the filtrate using boiling dichloromethane. Repeating this operation provided a third crop (60 mg). Overall yield: 690 mg, 1.93 mmol, 64%. ¹H NMR (CDCl₃, 300 MHz) δₕ 7.71 (dd, *J* = 7.6, 0.9 Hz, 2H), 7.61 (dd, *J* = 8.0, 0.9 Hz, 2H), 7.46 (app. t, *J* = 7.8 Hz, 2H). ¹³C{¹H} NMR (75 MHz, CDCl₃) δc 144.4, 139.3, 134.3, 133.8, 123.3, 121.3. HRMS (DCI-CH₄) Calc'd for $C_{12}H_7OS$ [M + H]⁺ 356.8584, found 356.8588. FTIR (neat) vmax/cm⁻¹ 3055, 2993, 1575, 1557, 1462, 1421, 1105, 1044, 1032, 1025, 787. Crystal data: see "Crystallographic data" section of the Supplementary Information file.

### Procedure for the photoreaction of (o-Br)₂-DBTO in solution

**(o-Br)₂-DBTO** (35.8 mg, 0.1 mmol) was dissolved in dry and freeze-pump-thaw degassed DCM (10 mL) under an inert atmosphere in a quartz Schlenk. The solution was then irradiated at room temperature using a UVA fluorescent tube (see Fig. S13 for the lamp emission spectrum) for 48 h at which point TLC analysis showed complete conversion. The volatiles were removed in vacuo and the crude mixture was then purified over silica gel chromatography (gradient from 0 to 50% DCM: Pentane) yielding successively **(o-Br)₂-DBT** (16 mg, 0.047 mmol, 47%) and **Br₂-OTO** (6.1 mg, 0.016 mmol, 16%). The missing yield in sulfur derivatives may arise from photobleaching and tar deposition observed on the quartz glassware after extended irradiation.

### Sample preparation and UV-irradiation conditions

The samples were prepared in-situ in a preparation chamber with a base pressure $p \leq 10^{-9}$ mbar. Au(111) sample was cleaned through several sputtering (Ar⁺, 1 kV, 1-2 µA) and annealing cycles (~800 K). Molecules were deposited on the surface using a commercial Kentax UHV-evaporator[45] that features three independent and separately controllable evaporation cells (quartz crucibles) and comes with an integrated shutter and a water-cooling system. **(o-Br)₂-DBTO** adsorption on the surface was performed by sublimation during 20 min at 394 K while maintaining the sample at RT. A quartz balance was used to calibrate the flux of molecules before deposition on the surface. Before the deposition of the molecules, fragments of a bulk NaCl(111) sample were loaded in a second crucible of the Kentax and sublimed at a rate of ~0.03 ML.min⁻¹ and a temperature of 730 K onto the Au(111) sample held at 373 K yielding a coverage with large NaCl islands showing a thickness up to three monolayers on the Au(111) surface.

**(o-Br)₂-DBTO** molecules adsorbed on the as-prepared sample were exposed to a 280 nm wavelength irradiation using a LED of 15 W/m² intensity. This was performed by irradiation of the sample inside the SPM head. The light spot is estimated to have a diameter of ~7 mm. During the entire process, the sample was maintained in scanning position at low temperature (~25 K). Several irradiation durations have been used: 10 min, 20 min, 30 min, 60 min, 90 min, 150 min, 190 min, 270 min and 390 min. Surface investigations and scanning probe measurements were performed after each irradiation process, along with a statistical analysis to support the qualitative findings derived from STM and nc-AFM images.

### STM and nc-AFM experiments

The studies were conducted using a commercial nc-AFM/STM (LT-SPM Infinity, from Scienta Omicron) operated by a Nanonis Control System (SPECS MIMEA control unit). The measurements were performed under ultrahigh vacuum conditions ($p \leq 10^{-10}$ mbar) at low temperature (9 K), using a qPlus sensor (oscillation amplitude $A_{min}$ ~ 50 pm, resonance frequency ~23 kHz, quality factor $Q$ ~ 50 000). Experimental images are either acquired in regular STM mode, i.e. while the nc-AFM mode is disengaged (no tip oscillation), or with the nc-AFM mode engaged. In the latter case, we talk about dynamic STM and the tunneling current regulation setpoint is then indicated as a mean tunneling current (due to the tip oscillation cycles) <$I_t$>. When the nc-AFM mode is engaged, topography (<$I_t$> regulated) and frequency shift ($\Delta f$) images are acquired simultaneously.

Experimental images were processed with the WSxM software[46].

## Computational methods

Density functional theory (DFT) calculations at the gas phase were carried out using B3LYP[47] hybrid functional and 6−311 ++ G(2 d,p) basis set implemented in Gaussian 16 software package[48]. The charge distribution was analyzed using the natural bond orbital method (NBO 7.0)[49].

## Data availability

The Supplementary Information file contains Supplementary Figures and Discussion, synthetic procedures and characterization of chemical compounds. Crystallographic data for the structures reported in this Article have been deposited at the Cambridge Crystallographic Data Centre, under deposition numbers CCDC-2366290 ((o-Br)$_2$-DBTO), CCDC-2366291 (Br$_2$-OTO) and CCDC-2366292 ((o-Br)$_2$-DBTO$_2$). Copies of the data can be obtained free of charge via https://www.ccdc.cam.ac.uk/structures/. The X-ray crystallographic coordinates for structures can be obtained free of charge from The Cambridge Crystallographic Data Centre via www.ccdc.cam.ac.uk/data_request/cif. Source Data are provided with this manuscript.

The remaining data generated in this study (SPM images, DFT models, NMR data, UV-vis spectroscopy data) have been deposited in the recherche.data.gouv.fr database under https://doi.org/10.57745/4LAQY0. All data can be obtained from the corresponding author upon request. Source data are provided with this paper.

## Code availability

The code developed in this study has been deposited in the recherche.data.gouv.fr database under https://doi.org/10.57745/4LAQY0.

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

## Acknowledgements

The authors thank the reviewers for their valuable and constructive comments about our work. This work was supported by the Centre National de la Recherche Scientifique (CNRS), Aix-Marseille Université and the Université de Toulouse. We thank the ANR funding agency for financial support of the CROSS project (ANR-21-CE09-0002, D.M., L.N. and C.K.) and the Light4Net project (ANR-21-CE09-0004, C.L. and L.N.). We thank Ms Iulia Cretoiu (LHFA) for her contribution to the synthesis of a sample of **(o-Br)₂-DBTO**, Franck Para (IM2NP) for experimental support and the services of the Institut de Chimie de Toulouse (ICT-UAR 2599) for their technical assistance.

## Author contributions

M.H.: formal analysis; investigation; validation; writing—original draft preparation; writing—review and editing. V.M.: data curation; investigation; validation; writing—review and editing. E.G.: formal analysis; investigation; software; validation; writing—review and editing. P.S.M.: investigation; writing - review and editing. S.C.: conceptualization; data curation; formal analysis; investigation; methodology; software; supervision; validation; writing - original draft preparation; writing—review and editing. L.G.: data curation; formal analysis; investigation; validation; writing—original draft preparation; writing—review and editing. C.L.: conceptualization; data curation; funding acquisition; formal analysis; investigation; methodology; supervision; validation; writing - original draft preparation; writing—review and editing. E.F.: formal analysis; investigation; software; validation; writing—review and editing. S.M.-L.: investigation; data curation; writing—review and editing. E.M.: conceptualization; supervision; validation; writing—review and editing. C.K.: conceptualization; data curation; funding acquisition; methodology; project administration; supervision; validation; writing—original draft preparation; writing—review and editing. D.M.: conceptualization; data curation; funding acquisition; methodology; project administration; supervision; validation; writing—review and editing. L.N.: conceptualization; data curation; formal analysis; funding acquisition; investigation; methodology; project administration; software; supervision; validation; writing—original draft preparation; writing—review and editing.

## Competing interests

The authors declare no competing interests.
