## [Transparent Peer Review file · Nature Communications]

Photoinduced Modulation of the Oxidation State of Dibenzothiophene S-Oxide Molecules on an Insulating Substrate

Corresponding Author: Dr Laurent Nony

Version 0:

Reviewer comments:

Reviewer #1

(Remarks to the Author)

In this manuscript, Hankache and co-workers report photoinduced deoxygenation on insulating alkali halide surfaces under ultra-high vacuum conditions. They use both scanning tunneling microscopy, non-contact atomic force microscopy, and bias spectroscopy to demonstrate that UV irradiation can promote the deoxygenation of dibenzothiophene S-oxide derivatives (DBTO) on a NaCl thin film formed on an Au(111). While similar photoinduced reactions of DBTO can be observed by solution chemistry, they claim to find a new type of on-surface reaction that can be transferred from in-solution reaction to on-surface synthesis. Although on-surface synthesis on an insulating substrate is of highly important regarding the synthesis of functional carbon-based materials, I feel current report here about photoinduced deoxygenation of DBTO is still in the early stage, and difficult to see any application prospects. I would like to recommend the publication in a specialized journal after the following issues have been well addressed.

1) More experimental evidence should be provided to support the photo-induced reaction from DBTO to DBT. Frequency shift and high-resolution STM image (see Fig. 4) can suggest a change of adsorbed DBTO, but to claim the chemical change from DBTO to DBT is experimentally insufficient. Control experiments, such as the STM image of adsorbed DBT (which can be prepared by in-solution chemistry) or bond-resolved nc-AFM images of both DBTO and DBT would be helpful.

2) Can photo-induced deoxygenation lead to a change in the packing structures of DBTO? Authors should compare the change of the unit cell parameters after UV irradiation.

3) The atomic directions marked in Fig. 2 ([010] and [100]) should be the non-polar directions of NaCl(001) rather than NaCl(100).

Reviewer #2

(Remarks to the Author)

The authors report on the on-surface photoinduced deoxygenation of the 2Br-DBTO precursor both in solution and on insulating NaCl thin layers/Au(111). This work reveals a new light-induced on-surface reaction on a decoupling surface, a very appealing topic due to the necessity of direct synthesis of carbon-based nanomaterials on semiconductor and insulating surfaces for technological applications. Furthermore, when comparing solution and on-surface synthesis a high chemo selectivity is observed, demonstrating the advantage of the latter bottom-up approach with respect to traditional synthesis. The article is concise and well-written. I find this work very interesting and it might be suitable for publication in nature communications. However, some issues should be addressed to meet the standards of the journal.

- Figure S4 shows a lateral modulation in the contrast of the molecules, probably from the corrugation observed on the NaCl/Au(111) originated from the herringbone reconstruction. Interestingly, this figure seems to reveal a strong dependence on the photoinduced deoxygenated molecules with respect to this modulation. To clarify this effect more overviews should be provided, and if so, the role of the Au(111) discussed to understand the mechanism triggering the reaction.

- The authors experimentally demonstrate the charge state of the sulfur atom by bias spectroscopy comparing the LCPD

between the pristine molecule and the photodeoxygenated molecule. However, to make a direct comparison more experimental information is necessary. First of all, it is not clear in the manuscript if all the parabolas were acquired at constant height or adjusting the tip-sample distance by $df(Z)$ spectroscopy. The latter information is also important to know the height of the sulfur atoms before and after illumination (the author point out that after illumination molecules flatten out on the surface due to deoxygenation), since the tip-sample distance is crucial to access information about charge distribution at the atomic level. The evolution of the parabolas (and LCPD) with respect to the tip height will support the conclusions of the manuscript.

Additional comments:

- The authors mention experiments performed on the bulk NaCl(100), experimental ncAFM data would help the comparison with experiments on NaCl(100) thin films on Au(111) and understand if the Au(111) is playing any role. Furthermore, did the authors perform similar illumination experiments on the bulk NaCl?
- In figure 2b the plane is not well subtracted.
- In Figure S3b the authors comment on the change in the Δf contrast around the sulfur atom after UV illumination as the molecules flatten out. However, the main feature observed is the interatomic interactions between consecutive molecules within the same row. Can the authors comment on the new interactions stabilizing the supramolecular phase after UV irradiation? These interactions can affect the charge state of the molecule (larger LCPD deviation is observed for deoxygenated molecules in figure S6)?
- Statistical analysis of the yield should involve more molecules to be representative.
- What was the yield after 10 and 70 minutes of illumination? Have the authors tried longer illumination times? Note that in solution the reaction requires 48 hours, and the yield is limited by the side-products while on surface is highly selective and higher yield can be achieved.
- Have the authors illuminated the sample at RT? Or anneal it to RT after illumination in the SPM head?

Reviewer #3

(Remarks to the Author)

The manuscript by Hankache and coworkers primarily investigates the photoinduced deoxygenation of dibenzothiophene S-oxide (DBTO) derivatives on insulating alkali halide surfaces. This research aims to expand the scope of 2D-confined surface reactions to improve in-solution reactions. Though I am more familiar with in-solution photochemistry, the prospects of controlling reaction geometries using surfaces present new opportunities and advantages over the in-solution methods for synthesis. Also, to my knowledge, this work represents the first time photodeoxygenation of an organic molecule has been investigated on a surface. Adding to the significance of this report is the apparent difference in the solution and surface photochemistry. The reported data generally supports the stated conclusions, and the highly technical experiments seem to have been expertly run. Thus, I would support the publication of this work after addressing a minor critique.

Critique: While the suggested in-solution mechanism is possible (lines 143-150), other possibilities need to be considered.

The first consideration is that photodeoxygenation photoproducts are often very sensitive to trace amounts of molecular oxygen. For example, while sultine (Br₂-OTO) is generally not observed for DBTO derivatives, selenium-analog has produced an analogous product depending on the amount of O₂ in the solution.(JACS 2004, pg 16058). Also, DBT derivatives have been shown to oxidize to DBTO₂ in the presence of a sensitizer.(JPCB, 2005, pg 8270). The report states the solutions were degassed and kept under an inert atmosphere but does not specify the technique by which the solutions were degassed. If the samples were sparged with inert gas, I recommend using freeze-pump-thaw techniques further to reduce the amount of O₂ in the solution and evaluate the effect on the observed photoproducts.

Another consideration that needs to be addressed by the manuscript is the potential for interconversion between Br₂-OTO and o-Br₂-DBTO₂. Along the same lines, was there any evidence for the o-Br₂-DBTO₂ undergoing photochemistry reported for the parent DBTO₂ (TetLett, 1994, pg 7155)

Ref 17 argues that the photodeoxygenation of DBTO releases singlet oxygen and not O(3P), as implied in line 86. However, most of the work after 1990 supports the release of O(3P), so I am not sure if Ref 17 needs to be included in this work.

While I believe the authors that no evidence for Br₂-OTO or o-Br₂-DBTO₂ was observed on the surface, a few sentences explaining how the data was used to reach this conclusion would have been appreciated by this non-expert.

This critique is a relatively minor point since the significant advance of this work is the surface photochemistry. However, since O₂ is almost surely absent in ultra-high vacuum conditions, I believe the manuscript should be rewritten to acknowledge the potential for O₂ to play a role in the solution chemistry or further investigation to rule out such a possibility.

Version 1:

Reviewer comments:

Reviewer #1

(Remarks to the Author)

My questions and concerns have been well addressed. I thus recommend for publication without further modifications.

Reviewer #2

(Remarks to the Author)

The manuscript has been well revised and additional experimental data were provided to support the conclusions. The manuscript is recommended for publication.

Reviewer #3

(Remarks to the Author)

The authors have added additional experiments that addressed my initial concerns. Additionally, they have updated the manuscript's text to address my other concerns.

Manuscript Title: Photoinduced Modulation of the Oxidation State of Dibenzothiophene S-Oxide Molecules on an Insulating Substrate

Authors: Mélissa Hankache *et al.*

Type of Article: Full Research Paper

Manuscript ID: NCOMMS-24-44100

Review Number: 1

The authors thank the referees for their meticulous reading of our work and their valuable comments. They have been explicitly thanked in the acknowledgments section of our manuscript. We have considered all of their comments in detail and provide complementary experiments in the revised version of our manuscript, which strengthen our initial conclusions. Our responses to all of their points are reported hereafter in detail **in red**.

REVIEWER COMMENTS

Reviewer #1 (Remarks to the Author):

In this manuscript, Hankache and co-workers report photoinduced deoxygenation on insulating alkali halide surfaces under ultra-high vacuum conditions. They use both scanning tunneling microscopy, non-contact atomic force microscopy, and bias spectroscopy to demonstrate that UV irradiation can promote the deoxygenation of dibenzothiophene S-oxide derivatives (DBTO) on a NaCl thin film formed on an Au(111). While similar photoinduced reactions of DBTO can be observed by solution chemistry, they claim to find a new type of on-surface reaction that can be transferred from in-solution reaction to on-surface synthesis. Although on-surface synthesis on an insulating substrate is of highly important regarding the synthesis of functional carbon-based materials, I feel current report here about photoinduced deoxygenation of DBTO is still in the early stage, and difficult to see any application prospects. I would like to recommend the publication in a specialized journal after the following issues have been well addressed.

1) More experimental evidence should be provided to support the photo-induced reaction from DBTO to DBT. Frequency shift and high-resolution STM image (see Fig. 4) can suggest a change of adsorbed DBTO, but to claim the chemical change from DBTO to DBT is experimentally insufficient. Control experiments, such as the STM image of adsorbed DBT (which can be prepared by in-solution chemistry) or bond-resolved nc-AFM images of both DBTO and DBT would be helpful.

Response from the authors:

We have carefully addressed that comment of the referee, as we partially do not agree with his statement. High-resolution nc-AFM imaging with a CO tip, as shown in Fig.S3b,

clearly indicates a change of contrast between pristine and deoxygenated molecules. To be more convincing, a finer analysis of Fig.S3b was performed (see Fig.S3c and S3d of the revised Supplementary Information file) that is fully consistent with the deoxygenation of the molecules. Specifically, it can be seen that the deoxygenated S atom is notably distinguishable from that of the pristine molecule as it exhibits a Δf that is -2 Hz lower than that of the pristine molecule, hence more attractive. This, combined with the rest of our results (bias spectroscopy experiments, DFT calculations, time evolution of the deoxygenation yield, spatial statistical analysis of the deoxygenation, see hereafter), makes our conclusion about the deoxygenation strongly robust. Regarding additional control experiments, performing bond-resolved nc-AFM images on such weakly adsorbed molecules turned out to be extremely delicate, either yielding disruption of the molecular domain due to the too-large short-range interaction regime, or making the tip unstable with our experimental conditions. Alternatively, we performed two distinct sets of additional measurements. On the one hand, we thoroughly investigated the statistics of the deoxygenation process, both spatially and in time. On the other hand, we performed additional bias spectroscopy measurements in grid mode over several molecules. All those new results point towards the same influence of UV irradiation on the deoxygenation of the molecules. To keep the main manuscript clear and concise, these additional measurements are detailed in the revised Supplementary Information file (see Fig.S4, S5, S6, S7 and S10).

2) Can photo-induced deoxygenation lead to a change in the packing structures of DBTO? Authors should compare the change of the unit cell parameters after UV irradiation.

Response from the authors:

Photo-induced deoxygenation does not lead to a change in the packing structures of DBT/DBTO. The unit cell parameters before/after irradiation remain unchanged. The question has been answered in Fig.3 (revised manuscript).

3) The atomic directions marked in Fig. 2 ([010] and [100]) should be the non-polar directions of NaCl(001) rather than NaCl(100).

Response from the authors:

This is correct. The numbering of the polar/non-polar directions has been revised to be consistent with the NaCl(100) plane.

Reviewer #2 (Remarks to the Author):

The authors report on the on-surface photoinduced deoxygenation of the 2Br-DBTO precursor both in solution and on insulating NaCl thin layers/Au(111). This work reveals a new light-induced on-surface reaction on a decoupling surface, a very appealing topic due to the necessity of direct synthesis of carbon-based nanomaterials on

semiconductor and insulating surfaces for technological applications. Furthermore, when comparing solution and on-surface synthesis a high chemo selectivity is observed, demonstrating the advantage of the latter bottom-up approach with respect to traditional synthesis. The article is concise and well-written. I find this work very interesting and it might be suitable for publication in nature communications. However, some issues should be addressed to meet the standards of the journal.

- Figure S4 shows a lateral modulation in the contrast of the molecules, probably from the corrugation observed on the NaCl/Au(111) originated from the herringbone reconstruction. Interestingly, this figure seems to reveal a strong dependence on the photoinduced deoxygenated molecules with respect to this modulation. To clarify this effect more overviews should be provided, and if so, the role of the Au(111) discussed to understand the mechanism triggering the reaction.

Response from the authors:

Based on Fig.S4, we do not see a strong dependence of the deoxygenated molecules with respect to the modulation of the Au(111) surface. However, the question is valuable, despite being difficult to address. The fact that the molecules align along the polar lines of the NaCl(100) substrate seems to reveal that their physical chemistry is likely to be influenced by that substrate rather than by Au(111). We therefore anticipate the deoxygenation process to be steered by the interactions between the molecules and the NaCl(100). However, we are aware that 2ML NaCl(100) are not enough to fully decouple the molecules from the influence of the conduction electrons of the Au(111) surface. As now discussed in the revised manuscript, we carried out a statistical study of the deoxygenation yield (deoxygenated molecules / pristine molecules) of the molecules on the surface as a function of sample illumination time. After each illumination phase, several images of around 450 molecules each were acquired from different areas of the sample, and the deoxygenation yield was estimated for each image. Those results are more detailed in another response to a comment from the referee (see below). The point here is that it was never observed that the Au(111) reconstruction, sometimes visible underneath the 2ML NaCl layers with our tunneling conditions, had an obvious influence on the location of the deoxygenated molecules. Besides, the influence that the pristine/deoxygenated state of neighboring molecules might have on the ability for a given molecule to become deoxygenated has also been investigated (Fig.S7). The statistical analysis shows that the state of neighboring molecules (all deoxygenated, none deoxygenated, or a mixture of both) has little influence on the ability of the central one to be deoxygenated. This is not a proof of the non-influence of the substrate on the deoxygenation, as its influence is not considered explicitly in the simulations, but since these reproduce quite well the experimental distribution of pristine/deoxygenated molecules, the substrate is expected, in first approximation, to play a minor role. We therefore conclude that the substrate (both NaCl(100) and Au(111)) has little influence, if any, on the spatial occurrence of the deoxygenation. As a consequence, the deoxygenation process seems to occur essentially randomly on the surface, whatever the adsorption spot and the pristine/deoxygenated state of the neighboring molecules.

- The authors experimentally demonstrate the charge state of the sulfur atom by bias spectroscopy comparing the LCPD between the pristine molecule and the photodeoxygenated molecule. However, to make a direct comparison more experimental information is necessary. First of all, it is not clear in the manuscript if all the parabolas were acquired at constant height or adjusting the tip-sample distance by $df(Z)$ spectroscopy. The latter information is also important to know the height of the sulfur atoms before and after illumination (the authors point out that after illumination molecules flatten out on the surface due to deoxygenation), since the tip-sample distance is crucial to access information about charge distribution at the atomic level. The evolution of the parabolas (and LCPD) with respect to the tip height will support the conclusions of the manuscript.

Response from the authors:

That comment is highly valuable. To better quantify the Δf due to the flattening of the molecules on the surface after deoxygenation, a finer analysis of Fig.S3b was performed. The deoxygenated S atom exhibits a Δf that is -2 Hz lower, hence more attractive, than that of the pristine molecule. However, the image being acquired at constant (mean) tunneling current ($\langle I_t \rangle$, dynamic STM), the distance at which the tip is located from each kind of S-atom depends on the local atomic orbital configuration, and is therefore difficult to be estimated in detail. Nevertheless, it is correct to state that the z-dependence of the CPD is crucial to access information about the charge distribution at the atomic level. As discussed in the manuscript, due to difficult tip conditions, the point spectroscopy experiments reported in Fig.4 have not been performed at constant height, but with constant tunneling feedback to set the reference tip-surface distance, yielding the trend reported in Fig.4b for the CPD. We anticipate the potential influence of the z-dependence of the CPD for these measurements to be weak. However, we did not perform additional z-dependent spectroscopic experiments, because they were found to be difficult on such weakly adsorbed molecules, with damaging risks of both, the tip and the sample. Instead, we performed true constant height grid spectroscopy measurements with a CO tip (see Supplementary Information file, section I.7b and Fig.S10). Despite different tip conditions compared to the first set of results, the Δf^* map is quite similar to that of the constant $\langle I_t \rangle$ image and the CPD above the deoxygenated S atom is also observed to be shifted towards larger values than that of pristine molecule (Fig.S10), meaning that the deoxygenated S atom carries a less positive charge compared its oxidized counterpart, as also expected from DFT calculations (Fig. S8). This states that for other tip conditions and other tip-surface distance conditions, the observed CPD trend is similar. Although the complete z-dependence of the CPD was not evidenced, we hope this additional experimental proof to be convincing enough to support the conclusions of the manuscript.

Additional comments:

- The authors mention experiments performed on the bulk NaCl(100), experimental ncAFM data would help the comparison with experiments on NaCl(100) thin films on

Au(111) and understand if the Au(111) is playing any role. Furthermore, did the authors perform similar illumination experiments on the bulk NaCl?

Response from the authors:

At room temperature, on bulk NaCl(100), (**o-Br**)₂-DBTO molecules tend to dewet quickly (~1h), which does not leave enough time to assess the influence of UV irradiation. Low-temperature experiments on bulk NaCl(100), being much more demanding than combined STM/nc-AFM experiments, have not been performed.

- In figure 2b the plane is not well subtracted.

Response from the authors:

Figure 2 has been changed. We have made sure that the planes of our experimental images are properly subtracted (Fig.2a,b,c, and 3a in the revised manuscript).

- In Figure S3b the authors comment on the change in the Δf contrast around the sulfur atom after UV illumination as the molecules flatten out. However, the main feature observed is the interatomic interactions between consecutive molecules within the same row. Can the authors comment on the new interactions stabilizing the supramolecular phase after UV irradiation? These interactions can affect the charge state of the molecule (larger LCPD deviation is observed for deoxygenated molecules in figure S6)?

Response from the authors:

This is another highly valuable comment. Indeed, as explained above, in Fig.S3b, the UV illumination induces the flattening of the deoxygenated molecule. This translates by a -2Hz change of the Δf measured above the deoxygenated S atom compared to that of the pristine molecule. But it can also be noticed in that figure that the carbon backbone of the molecule next to the deoxygenated one within the same row is slightly lifted up (brighter parts in Fig.S3b) compared to other pristine molecules. We therefore think that the conformational change of the deoxygenated molecule favors a stronger adsorption w.r.t. the NaCl(100), while reinforcing the H...Br and H...S interactions between adjacent molecules. The optimization of the regioselectivity of these interactions seems to force the carbon backbone of the pristine molecule to be slightly lifted. Finally, we would like to emphasize two points: i- as stated in the manuscript, we have estimated the molecular unit cell before illumination (rectangular cell, 2 molecules per cell: $a_m \times b_m = 6a \times 2a$, where a is the centered unit cell of NaCl crystal) and after UV illumination. It turns out that the unit cells of the molecular domains remain unchanged before and after illumination. Thus, the local structural change induced by the deoxygenation of some molecules within the domains does not affect their epitaxy on NaCl(100), ii- The CPD map in Fig.S10, that may be interpreted to first order as a charge state analysis of the molecules, reveals that the two deoxygenated molecules (in the center and on the right) have an overall larger CPD compared to that of the pristine molecule (on the left). These molecules have a different charge state compared to the pristine ones (they are less positive, or more negative). So, the referee is right, the interactions between pristine and deoxygenated molecules can affect both their

charge distribution and their local adsorption configuration, without however affecting the overall structure of the molecular domains.

- Statistical analysis of the yield should involve more molecules to be representative.

Response from the authors:

This concern has been addressed in detail (see response above and below).

- What was the yield after 10 and 70 minutes of illumination? Have the authors tried longer illumination times? Note that in solution the reaction requires 48 hours, and the yield is limited by the side-products while on surface is highly selective and higher yield can be achieved.

Response from the authors:

As now discussed in the revised manuscript and above, we carried out a statistical study of the deoxygenation yield of molecules on the surface as a function of sample illumination time (Fig.S4). After each illumination phase, several images of around 450 molecules were acquired from different areas of the sample, and the deoxygenation yield was estimated for each image by two separate methods, both of which gave the same yield estimate (Fig.S5 and S6). It is found that the deoxygenation yield increases rapidly with illumination time (0-30% in 50 minutes), then reaches saturation at 40%, after more than six hours of illumination (Fig.S4). The origin of the saturation was not investigated further in detail in this work. Long exposure time yields to an increase in the temperature of the SPM head (~25K). Owing to the presence of the saturation of the deoxygenation yield and in order to preserve the sample quality, we did not investigate longer exposure times.

- Have the authors illuminated the sample at RT? Or anneal it to RT after illumination in the SPM head?

Response from the authors:

The illumination and the SPM investigations reported here have all been performed at 10K. Illuminations at RT have not been performed as the molecules tend to quickly dewet on the surface at that temperature. Similar investigations cannot therefore be performed at RT.

Reviewer #3 (Remarks to the Author):

The manuscript by Hankache and coworkers primarily investigates the photoinduced deoxygenation of dibenzothiophene S-oxide (DBTO) derivatives on insulating alkali halide surfaces. This research aims to expand the scope of 2D-confined surface reactions to improve in-solution reactions. Though I am more familiar with in-solution photochemistry, the prospects of controlling reaction geometries using surfaces present new opportunities and advantages over the in-solution methods for synthesis. Also, to my knowledge, this work represents the first time photodeoxygenation of an organic molecule has been investigated on a surface. Adding to the significance of this report is the apparent difference in the solution and surface photochemistry. The reported data generally supports the stated conclusions, and the highly technical experiments seem to have been expertly run. Thus, I would support the publication of this work after addressing a minor critique.

Critique: While the suggested in-solution mechanism is possible (lines 143-150), other possibilities need to be considered. The first consideration is that photodeoxygenation photoproducts are often very sensitive to trace amounts of molecular oxygen. For example, while sultine ($\text{Br}_2\text{-OTO}$) is generally not observed for DBTO derivatives, selenium-analog has produced an analogous product depending on the amount of O_2 in the solution.(JACS 2004, pg 16058). Also, DBT derivatives have been shown to oxidize to DBTO_2 in the presence of a sensitizer.(JPCB, 2005, pg 8270). The report states the solutions were degassed and kept under an inert atmosphere but does not specify the technique by which the solutions were degassed. If the samples were sparged with inert gas, I recommend using freeze-pump-thaw techniques further to reduce the amount of O_2 in the solution and evaluate the effect on the observed photoproducts.

Another consideration that needs to be addressed by the manuscript is the potential for interconversion between $\text{Br}_2\text{-OTO}$ and $o\text{-Br}_2\text{-DBTO}_2$. Along the same lines, was there any evidence for the $o\text{-Br}_2\text{-DBTO}_2$ undergoing photochemistry reported for the parent DBTO₂ (TetLett, 1994, pg 7155)

Response from the authors:

Following the reviewer's suggestion, we have reproduced the in-solution photoreaction using dichloromethane degassed by freeze-pump-thaw cycles. Under such conditions, photoirradiation of $(o\text{-Br})_2\text{-DBTO}$ gave $(o\text{-Br})_2\text{-DBT}$ in 47% isolated yield, along with the sultine $\text{Br}_2\text{-OTO}$ (16%) as sole byproduct. The missing yield may arise from photobleaching and tar deposition observed on the quartz glassware after extended irradiation. Importantly, no trace of the sulfone $(o\text{-Br})_2\text{-DBTO}_2$ was observed, contrary to the conditions reported in the initial version of manuscript, employing argon sparged dichloromethane as solvent.

The photoreactivity of sultine $\text{Br}_2\text{-OTO}$ and sulfone $(o\text{-Br})_2\text{-DBTO}_2$ were also investigated. Photoirradiation (UVA) of a dichloromethane solution of pure sultine $\text{Br}_2\text{-OTO}$ did not lead to any conversion. Likewise, when sulfone $(o\text{-Br})_2\text{-DBTO}_2$ was photoirradiated (UVA), no conversion of the starting material was observed, and in particular, no trace of the corresponding dibromobiphenyl product was detected.

These experiments therefore confirm that there is no interconversion between sultine **Br₂-OTO** and sulfone **(o-Br)₂-DBTO₂**, and that the sulfone is not formed under thoroughly deoxygenated conditions. This points towards two distinct mechanisms for the formation of the sultine and the sulfone. According to our results, we hypothesize that the sulfone may be formed by side reactions involving molecular oxygen, while the sultine may result from an intermolecular transfer of atomic oxygen, presumably involving a C-S α -cleavage.

In the revised manuscript, we have now reported the results of the photodeoxygenation reaction in thoroughly degassed dichloromethane (by freeze-pump-thaw cycles), so as to exclude any contribution of adventitious molecular oxygen, that is absent under the ultra-high vacuum conditions used for on-surface photoreactions. We have thus modified the text and Figure 1 accordingly.

In the revised Supplementary Information file, we have updated the caption of Figure S1 with the outcome of the in-solution reaction under thoroughly deoxygenated conditions. In addition, in section II.3 describing the photoreactivity in solution, we have detailed the procedure and the outcome of the photodeoxygenation reaction using degassed dichloromethane obtained by argon sparging and by freeze-pump-thaw cycles (paragraphs a and b of section II.3). Finally, a comment on the absence of photoreactivity of sultine **Br₂-OTO** and sulfone **(o-Br)₂-DBTO₂** under the reaction conditions was added (paragraph d of section II.3).

Ref 17 argues that the photodeoxygenation of DBTO releases singlet oxygen and not O(3P), as implied in line 86. However, most of the work after 1990 supports the release of O(3P), so I am not sure if Ref 17 needs to be included in this work.

Response from the authors:

We fully agree with the reviewer that this primary literature reference from 1973 reporting photochemical deoxygenation of aryl sulfoxides supports a reaction mechanism that is different from the now well-established release of O(3P) by S-O bond cleavage. Therefore, for the sake of clarity to readers that may not be specialists in the field, we have now suppressed reference 17 from the manuscript.

While I believe the authors that no evidence for Br₂-OTO or o-Br₂-DBTO₂ was observed on the surface, a few sentences explaining how the data was used to reach this conclusion would have been appreciated by this non-expert.

Response from the authors:

We thank the reviewer for this suggestion that will clearly help the readers that are non-experts in Scanning Probe Microscopy techniques to understand how conclusions are drawn for the on-surface photoreactivity experiments.

The absence of sulfone **(o-Br)₂-DBTO₂** and sultine **Br₂-OTO** after on-surface photoirradiation was established according to experimental images acquired by Scanning Tunneling Microscopy and/or non-contact Atomic Force Microscopy. In particular, the latter allows to distinguish on the recorded images the backbone of organic compounds, as can be seen on Fig. S3b and S3c (revised Supplementary

Information file) for **(o-Br)₂-DBT** and **(o-Br)₂-DBTO**, for which the central five-membered ring can be clearly visualized. The presence of a six-membered ring as central core (*i.e.* of a phenanthrene-derived heteroaromatic scaffold) has never been observed experimentally in the investigated population of molecules, which allows us to conclude that the **Br₂-OTO** sulfone is not formed during on-surface photochemistry experiments.

In addition, to assign the structure (and the on-surface adsorption configuration) of **(o-Br)₂-DBT** derivatives all containing a central five-membered ring, a comparison of experimental images with simulated ones, obtained with the Particle Probe Model (Fig. S1), was undertaken. The contrast related to one oxygen pointing “up” with regard to the surface (Fig. S1c and S1d) was never observed experimentally, thus allowing to assign the adsorption configuration of **(o-Br)₂-DBTO** on the surface and most importantly here, to exclude the photoinduced formation of the **(o-Br)₂-DBTO₂** sulfone (which contains both one oxygen pointing “down” and one oxygen pointing “up”).

In the revised manuscript, a sentence stating that the **Br₂-OTO** sulfone scaffold has never been observed on surface after photoirradiation experiments has been added. In the revised Supplementary Information file, the text accompanying Figure S1 has been completed, in order to help the readers that are non-experts in Scanning Probe Microscopy techniques to understand how conclusions are drawn regarding on the absence of **Br₂-OTO** and **(o-Br)₂-DBTO₂** on surface.

This critique is a relatively minor point since the significant advance of this work is the surface photochemistry. However, since O₂ is almost surely absent in ultra-high vacuum conditions, I believe the manuscript should be rewritten to acknowledge the potential for O₂ to play a role in the solution chemistry or further investigation to rule out such a possibility.

Response from the authors:

Indeed, the new photoreactivity experiment carried out in thoroughly degassed DCM, using freeze-pump-thaw technique, allowed to evidence the role of adventitious molecular oxygen in the formation, in solution, of the sulfone **(o-Br)₂-DBTO₂**.

This experiment also highlighted that, even under strictly deoxygenated conditions, the photoreaction of **(o-Br)₂-DBTO** in solution is not fully chemoselective and yields the sulfone **Br₂-OTO** as a byproduct, presumably due to atomic oxygen transfer. In contrast, this phenomenon is not observed in photoreactions carried out on surface under Ultra-High Vacuum conditions, thereby allowing for a full selectivity towards the reduced **(o-Br)₂-DBT**.